# Impella versus Venoarterial Extracorporeal Membrane Oxygenation for Acute Myocardial Infarction Cardiogenic Shock: A Systematic Review and Meta-Analysis

**DOI:** 10.3390/jcm11143955

**Published:** 2022-07-07

**Authors:** Riley J. Batchelor, Andrew Wheelahan, Wayne C. Zheng, Dion Stub, Yang Yang, William Chan

**Affiliations:** 1Department of Cardiology, The Alfred Hospital, 55 Commercial Road, Melbourne 3004, Australia; r.batchelor@alfred.org.au (R.J.B.); wa.zheng@alfred.org.au (W.C.Z.); d.stub@alfred.org.au (D.S.); 2Department of Cardiology, The Royal Melbourne Hospital, Melbourne 3004, Australia; 3Department of Cardiology, Western Health, Melbourne 3004, Australia; andrew.wheelahan@gmail.com; 4Department of Epidemiology and Preventive Medicine, Monash University, Melbourne 3004, Australia; 5Intensive Care Unit, Western Health, Melbourne 3004, Australia; yang.yang@wh.org.au; 6Department of Medicine, University of Melbourne, Melbourne 3052, Australia

**Keywords:** acute myocardial infarction, cardiogenic shock, VA-ECMO, Impella, mechanical circulatory support

## Abstract

Objectives: Despite an increase in the use of mechanical circulatory support (MCS) devices for acute myocardial infarction cardiogenic shock (AMI-CS), there is currently no randomised data directly comparing the use of Impella and veno-arterial extra-corporeal membrane oxygenation (VA-ECMO). Methods: Electronic databases of MEDLINE, EMBASE and CENTRAL were systematically searched in November 2021. Studies directly comparing the use of Impella (CP, 2.5 or 5.0) with VA-ECMO for AMI-CS were included. Studies examining other modalities of MCS, or other causes of cardiogenic shock, were excluded. The primary outcome was in-hospital mortality. Results: No randomised trials comparing VA-ECMO to Impella in patients with AMI-CS were identified. Six cohort studies (five retrospective and one prospective) were included for systematic review. All studies, including 7093 patients, were included in meta-analysis. Five studies reported in-hospital mortality, which, when pooled, was 42.4% in the Impella group versus 50.1% in the VA-ECMO group. Impella support for AMI-CS was associated with an 11% relative risk reduction in in-hospital mortality compared to VA-ECMO (risk ratio 0.89; 95% CI 0.83–0.96, I^2^ 0%). Of the six studies, three studies also adjusted outcome measures via propensity-score matching with reported reductions in in-hospital mortality with Impella compared to VA-ECMO (risk ratio 0.72; 95% CI 0.59–0.86, I^2^ 35%). Pooled analysis of five studies with 6- or 12-month mortality data reported a 14% risk reduction with Impella over the medium-to-long-term (risk ratio 0.86; 95% CI 0.76–0.97, I^2^ 0%). Conclusions: There is no high-level evidence comparing VA-ECMO and Impella in AMI-CS. In available observation studies, MCS with Impella was associated with a reduced risk of in-hospital and medium-term mortality as compared to VA-ECMO.

## 1. Introduction

Cardiogenic shock (CS) complicates acute myocardial infarction (AMI) in 5–10% of cases [1,2,3], and remains the leading cause of mortality in patients with AMI. CS is associated with mortality rates of 40–50% [4], in spite of timely reperfusion and improved medical therapy over time [3,5,6,7]. In addition to revascularisation, maintenance of adequate end-organ perfusion to prevent irreversible ischaemia and multi-organ failure is becoming increasingly recognised as a key target in patients with AMI-CS [8,9]. Where inotropic/vasopressor therapy is inadequate, haemodynamics may be improved with the use of mechanical circulatory support (MCS) devices. However, clinical trials and meta-analyses are yet to identify a difference in mortality with commonly utilised MCS devices, such as the intra-aortic balloon pump (IABP), as compared to best medical therapy, with insufficient supplementary cardiac output a potential reason for inadequacy [10,11]. While several more potent MCS alternatives have seen an increase in uptake and use over the past decade [12,13], there remains a paucity of randomised evidence for axial flow pump left ventricular assist devices (i.e., Impella), nor for veno-arterial extracorporeal membrane oxygenation (VA-ECMO) over standard therapy in patients with AMI-CS. Impella is a continuous flow pump, inserted either surgically (5.0) or percutaneously (2.5 and CP), which extracts blood from the left ventricle with active propulsion into the ascending aorta to decompress the left ventricle and provide forward cardiac output. Impella is commonly indicated in the treatment of AMI-CS due to predominant left ventricular failure where its therapeutic effects include mechanically unloading the left ventricle, improving forward cardiac output to sustain perfusion to vital organs, reducing pulmonary congestion and ameliorating pulmonary edema, and augmenting coronary artery perfusion to relieve myocardial ischemia. Impella support versus IABP has previously been evaluated in randomised controlled trials, with no statistically significant mortality difference in the short- and long-term, although the sample sizes were small [14,15]. VA-ECMO is an alternative high-output MCS modality, classically indicated in AMI-CS [16], but additionally provides therapeutic benefits among those with circulatory and respiratory collapse due to cardiac and/or respiratory arrest or those with more advanced vital organ dysfunction [17]. To date, there have been no randomised trials comparing Impella and VA-ECMO in AMI-CS [18]. Therefore, this systematic review and meta-analysis sought to evaluate the use of Impella versus VA-ECMO, as two alternative high-output forms of MCS in AMI-CS.

## 2. Methods

The methodology of this systematic review and meta-analysis was guided by the set of items outlined by the Preferred reporting Items for Systematic Reviews and Meta-Analyses: the PRISMA statement. A systematic search of the electronic databases MEDLINE, EMBASE and Cochrane Central Register of Controlled Trials (CENTRAL) was undertaken, identifying articles from inception to the search date in November 2021. A broad search strategy was utilised, encompassing medical subject headings and controlled search terms relating both specifically to the study interventions (i.e., Impella, VA-ECMO) and more generally (i.e., percutaneous left ventricular assist device (pLVAD), MCS, AMI-CS). The primary outcome was in-hospital mortality, and secondary outcomes were 6- and 12-month mortality. The search string was limited to the inclusion of human participants. The search strategy was adapted for each database. Editorial articles, letters, and other non-primary studies, including abstracts, were also screened for references to potentially relevant studies that may have been missed. A sample MEDLINE search strategy is included in Appendix A. Patients or the public were not involved in the design, conduct, reporting, or dissemination plans of this research.

### 2.1. Study Selection

All original peer reviewed studies involving patients over the age of 18 were included. Studies were included if they reported on patients with AMI-CS who required MCS support with VA-ECMO or Impella or both, with analysis based on the first device used only. Studies that compared MCS devices other than VA-ECMO or Impella (e.g., IABP, durable left ventricular devices) were excluded. Studies reporting on patient cohorts presenting with non-AMI causes of cardiogenic shock were excluded. Studies reporting on planned Impella or VA-ECMO support during high-risk elective percutaneous coronary intervention (PCI) were excluded. Randomized controlled trials and observational studies were included; however, case reports were excluded. Studies that could not be retrieved in full text form, such as abstracts, were excluded.

All studies identified were imported into an online review platform (Covidence) [19] and most duplicates automatically identified and deleted. Two review authors (RB and AW) independently completed title and abstract screening, with the same two authors then independently assessing the full texts of all studies that potentially contained relevant data. A senior author (WC) mediated any discordance during independent assessments.

### 2.2. Data Extraction

A pre-piloted data extraction form based on the minimum requirements recommended in the Cochrane Handbook for Systematic Reviews was used to collect relevant qualitative and quantitative data from included studies [20]. Two review authors (RB and AW) independently extracted data, with any discrepancy on comparison reviewed and mediated by discussion with a third author (WC).

### 2.3. Risk of Bias Analysis

The Newcastle–Ottawa Scale [21] was used to appraise observational data. Quality and risk of bias assessment was conducted independently by two review authors (RB and WZ), with discordance in assessment resolved by a third author (WC).

### 2.4. Data Synthesis and Statistical Analysis

Descriptive characteristics and measures of association from included studies were pooled, and estimates of effect were generated using Review Manager 5.3 [22] software. Where sufficient patient level data was not available (for example, cohort percentages with no number of patients outlined), conservative estimates were made based on the data available. Six-month mortality data was annualised and pooled with studies reporting mortality at one year, as to enable more robust analysis. The weighted mean, 95% confidence interval and *p*-value were generated and shown graphically in forest plots for exposure-outcome comparisons. The I^2^ statistic was used to determine the degree to which the effect estimate varied. An inverse variance method with random-effects modelling was used, given the methodological heterogeneity across included studies. Sensitivity and subgroup analyses were conducted following the main analysis to identify studies, or groups of studies, that influenced the result of the primary analysis (in-hospital mortality).

## 3. Results

The systematic search identified 1849 unique titles and abstracts, of which 1738 were excluded based on relevance to this review. The remaining 111 studies underwent full text screening, following which a further 105 articles were excluded, leaving six studies included [23,24,25,26,27,28]. Studies were mostly excluded based on population and outcome relevance. A PRISMA flow diagram outlining the screening process and all reasons for study exclusion is presented in Figure 1. The risk of bias assessment for all studies is presented in Appendix A. In general, the six studies were of reasonable to good methodological quality. A summary funnel plot was generated to assess publication bias that may have been present across studies reporting in-hospital mortality, with asymmetry testing performed suggestive of low risk (Appendix A).

### 3.1. Qualitative Analysis

No randomised controlled trials comparing VA-ECMO and Impella in AMI-CS were identified. The six articles included consisted of five retrospective cohort studies [23,25,26,27,28] and one prospective cohort study [24]. In total, the included observational studies accounted for 7093, with 818 patients supported on VA-ECMO, and 6275 patients on Impella. Five studies included patients that received Impella CP [23,24,26,27,28], four on Impella 2.5 [23,26,27,28], and two on Impella 5.0 [23,26]. Five studies reported on in-hospital mortality [24,25,26,27,28], and five studies reported on mortality at the combined 6- or 12- month time endpoint [23,24,25,27,28]. A detailed summary of study characteristics and relevant data is included in Table 1.

The largest observational study included in this systematic review, conducted in the United States by Lemor et al., utilised a national admissions database and ICD-10-CM codes to evaluate an adult population of 6290 patients with AMI-CS supported by VA-ECMO or Impella (CP, 2.5, 5.0), between October 2015 and December 2017 [26]. Patients that received both VA-ECMO and Impella, did not undergo PCI, or had missing mortality data were excluded. After adjustment for several covariates, including age, sex, and comorbidities, in-hospital mortality was found to be significantly higher in the VA-ECMO cohort as compared to Impella (45.5% versus 41.5%, adjusted OR 2.34, 95% CI 1.31–4.14, *p* = 0.004), with a higher rate of stroke, vascular complication, and bleeding in the VA-ECMO cohort [26]. A propensity score-matched cohort of 900 admissions (450 in each group) corroborated this finding (43.3% versus 26.7%, OR 2.10, 95% CI 1.12–3.95, *p* = 0.021).

Two retrospective cohort studies from Germany were included in this systematic review and meta-analysis, one from September 2014–September 2019, assessing patients with cardiogenic shock (83% AMI-CS in the VA-ECMO cohort and 87% AMI-CS in the Impella cohort) [27], and one assessing AMI-CS patients that presented with out-of-hospital-cardiac arrest (OHCA) [28]. Differing from the Lemor et al. study, these two studies utilised medical record data and did not exclude patients that received both devices, instead analysing based on first device used. Of patients with cardiogenic shock included in the Karatolios et al. study, 25.7% of patients in the Impella cohort had presented with OHCA compared to 40.6% of the VA-ECMO cohort, raising the likelihood of cohort overlap. Both studies adjusted for age, various comorbidities, time to return of spontaneous circulation, and measures of in-hospital progress, such as the vasoactive score. Both studies concluded that Impella (CP or 2.5) was associated with comparable in-hospital and 6- (Karatolios et al.) or 12- month (Syntila et al.) mortality [27,28].

The remaining three studies were comparably smaller, reported unadjusted measures of effect, and reported heterogenous exclusion criteria. The single prospective cohort study included 51 patients and stratified patients by first used device, reporting comparable in-hospital mortality between Impella and VA-ECMO (45% versus 45%) [24].

### 3.2. Meta-Analysis of in-Hospital and 6- or 12-Month All-Cause Mortality

Five observational studies reported in-hospital mortality in patients supported with Impella versus VA-ECMO [24,25,26,27,28]. Univariate data was available on 7051 participants and was used to generate a pooled effect estimate. The univariate overall risk ratio of in-hospital all-cause mortality with Impella versus VA-ECMO was 0.89 (95% CI 0.83–0.96, *p* = 0.004, I^2^ 0%) (Figure 2). Five observational studies reported all-cause mortality at 6–12-months [23,24,25,27,28]. Univariate meta-analysis included 803 participants, with the overall risk ratio of 6–12-month all-cause mortality with Impella versus VA-ECMO being 0.86 (95% CI 0.76–0.97, *p* = 0.02, I^2^ 0%) (Figure 3).

### 3.3. Sensitivity and Subgroup Analysis

Given the overlapping study population of the two large German retrospective cohort studies, secondary analyses limiting meta-analysis of all-cause mortality in-hospital and at 6- or 12- months were conducted, alternating exclusion of the Karatolios [27] and Syntila cohorts [28]. These sensitivity analyses were consistent with the primary analysis (Appendix A). Limiting meta-analysis to observational studies with the lowest risk of bias (New Castle Ottawa scale score ≥ 8) was also consistent with the primary analysis (in-hospital all-cause mortality, *n* = 4 studies, RR 0.89, 95% CI 0.82–0.96, I^2^ 0%; 6–12 month mortality, *n* = 3 studies, RR 0.87, 95% CI 0.77–0.98, I^2^ 0%). With respect to the primary outcome, removal of any single study from analysis did not result in a deviation in the significance of the data. As the primary analysis reported unadjusted measures of effect only, a separate subgroup analysis of propensity-matched cohorts was also conducted, demonstrating consistency with the primary findings with respect to in-hospital mortality (RR 0.72, 95% CI 0.59–0.86, I^2^ 35%) but not 6- or 12- month mortality (RR 0.88, 95% CI 0.72–1.09, I^2^ 0%) (Appendix A).

## 4. Discussions

The results of this systematic review of six observational studies suggest that there remains a paucity of high-quality evidence comparing the most widely used mechanical support devices in AMI-CS. Observational data suggests that Impella support is comparable to VA-ECMO in AMI-CS, and may be associated with a reduction in short- and medium-term all-cause mortality. Meta-analysis of 7051 patients across five studies found a reduction in in-hospital mortality of 11% in those treated with Impella. Meta-analysis of five observational studies that reported 6-month (annualised), or 12-month mortality data, including 803 patients, found a reduction in medium-term mortality of 14% with Impella support. While these results were derived from univariate observational data, they were robust to sensitivity analyses examining propensity-matched cohorts and examining cohort studies at low risk of bias and without population overlap. In the absence of robust randomized-controlled trial data directly comparing Impella versus VA-ECMO for clinical outcomes, results from this study underscore that both MCS modalities appear appropriate to support AMI-CS, but that Impella might be more well-suited physiologically with modest reduction observed in short- and medium-term mortality and complication rates.

This is the first systematic review and meta-analysis to directly compare Impella with VA-ECMO in AMI-CS. In patients hospitalised with AMI, CS complicates 5–8% of ST-elevation myocardial infarction (STEMI) and 3% of non-STEMI presentations, [29] and carries a poor prognosis, with in-hospital mortality rates between 40–50% [3,4,5,6,7]. In AMI-CS, myocardial ischaemia results in regional necrosis and a reduction in cardiac contractility and cardiac output, leading to systemic hypoperfusion, and activation of neurohormonal cascade resulting in vasoconstriction and fluid retention. This, in turn, increases myocardial afterload, which reduces cardiac indices further and leads to pulmonary congestion, pulmonary hypertension, and circulatory failure, often exacerbated by development of a systemic inflammatory response [3]. Intravenous vasoactive and inotropic therapy is often utilised as the first-line treatment to maintain vital end-organ perfusion, but is at the expense of an increase in myocardial workload and oxygen demand. When pharmacotherapy requirements escalate in AMI-CS, mortality has been shown to increase commensurately with the number of pharmacologic agents that are needed [30]. Therefore, consideration of early MCS in AMI-CS serves multiple purposes, including providing myocardial/left ventricular support, improving systemic haemodynamics, and improving systemic oxygenation to mitigate the negative spiral of AMI-CS [8].

VA-ECMO can, in all forms of shock, provide haemodynamic support and maintain end-organ perfusion as a bridge to destination therapy or recovery. However, specific to the case of AMI-CS, VA-ECMO, by virtue of the femoral arterial cannula and retrograde supply of oxygenated blood, increases left ventricular afterload, potentially resulting in left ventricular distension and refractory pulmonary oedema unless left ventricular unloading takes place. While IABP counterpulsation was, for many years, considered first line management in AMI-CS, the IABP-SHOCK II trial in 2012 demonstrated no significant difference in 30-day mortality with IABP versus standard medical therapy, and furthermore highlighted the importance of rigorously performed randomised controlled trials to guide decision making in this difficult group of patients [10]. Percutaneous left ventricular assist devices, such as the Impella family of devices, are continuous flow, micro-axial pumps seated within the left ventricle, which eject blood from the pump into the ascending aorta. From a physiology standpoint, Impella decompresses the left ventricle, reduces pulmonary congestion, improves forward cardiac output, and improves myocardial oxygenation by augmenting coronary perfusion pressure [31,32,33]. While Impella has been previously compared to IABP in two small, randomised trials (*n* < 100 patients total) with neither trial demonstrating mortality benefit over IABP, both the studies were significantly underpowered for mortality endpoint [14,15].

There are no randomised trials evaluating the use of Impella versus VA-ECMO in AMI-CS. Cohort studies time frames included in this systematic review ranged between the earliest year in 2006 and the latest in 2020. As identified in qualitative analysis, existing observational data is heterogenous, with the number of patients on Impella CP, 2.5, and 5.0 varying between each study. The largest observational study included in this systematic review derived data from ICD-10 codes and national database data [26], whereas the other five smaller studies interrogated medical records. Three studies excluded patients that either had MCS commenced during ongoing cardiopulmonary resuscitation or refractory arrest, or those that had cardiac arrest without return of spontaneous circulation [23,25,27]. The largest three retrospective cohort studies reported propensity-matched in-hospital mortality, and adjusted for a series of relevant multivariate factors, including age, Charlson comorbidity index, and vasoactive score [26,27,28]. Not all risk factors known to be relevant to mortality in acute coronary syndrome were reliably adjusted for, such as downtime and bystander CPR in those with out-of-hospital arrest, lesion complexity, left ventricular function, and renal function [34,35]. The percentage of patients that presented with cardiac arrest was heterogenous across studies, which is relevant as the experience with MCS in high volume shock centres, where all devices are available, is likely to depend on operator expertise and preference based on patient presentation and demographics which may also alter outcomes, for example, VA-ECMO for AMI-CS with cardiac arrest, and Impella for CS identified while already undertaking percutaneous coronary intervention. Regardless, the propensity score-matched cohorts appeared to mirror the primary adjusted findings in these cohort studies, all favouring the use of Impella over VA-ECMO in AMI-CS. Generally, the VA-ECMO cohort had higher rates of major bleeding, thromboembolic events, and peripheral limb ischaemia than the Impella cohort (Table 1), which may represent another mechanism by which the lower associated mortality in the Impella group is supported. Irrespective of means of temporary MCS, in-hospital mortality with AMI-CS was still high, ranging from 40–60% across included studies, highlighting the acuity with which these patients present.

This study adopted a comprehensive approach to reviewing the current literature on VA-ECMO and Impella use in AMI-CS, utilising the Cochrane methodology in study screening, selection, appraisal, and data extraction. The results of our study should be interpreted in the context of likely significant selection bias for the type of MCS used to support AMI-CS. As there are currently no randomized-controlled trials evaluating different modalities of MCS for CS, the included observational studies would be subject to inherent selection bias for MCS choice influenced by, but not limited to, factors including physician familiarity of the MCS type, institutional availability of the MCS type, severity of CS, the associated vital organ dysfunction (such as predominance of right ventricular failure, and respiratory failure), as well as concomitant cardiac arrest or unstable ventricular arrythmias. A single cohort study of 6290 patients utilising coding data was heavily weighted given the relatively smaller size of other studies included in meta-analysis. Despite low I2 statistical values, likely due to the use of unadjusted cohort data in a random effects model, study designs were variable, with heterogenous exclusion criteria and significant clinical diversity. AMI-CS itself is a heterogenous condition, with left and right ventricular failure, mechanical complications, and arrythmias all contributing. VA-ECMO itself is often used in more severe forms of shock, and thus may confer higher mortality than would otherwise have been expected. Raw univariate data was utilised for primary analyses as not all studies adjusted for significant covariates. Even though the primary analysis was robust to sensitivity analysis and mirrored findings when restricted to propensity-matched data, no measure of causality can be drawn from an analysis of longitudinal data of this nature.

In conclusion, the results of this systematic review and meta-analysis of six cohort studies comparing VA-ECMO to Impella suggest that both MCS modalities appear appropriate to support AMI-CS, but that Impella may be more well-suited for this subset of cardiogenic shock patients, with modest reduction observed in short- and medium-term mortality and complication rates. The studies included in this review were heterogenous in study design, which creates a challenge when implementing the findings of this systematic review in the real world, given that many centres may only have one MCS device available. Further evaluation of the findings of this review, and assessment of causality, requires the engagement of robust randomized-controlled trials powered for clinical endpoints, of which there are currently none, but they are necessary.

## Figures and Tables

**Figure 1 jcm-11-03955-f001:**
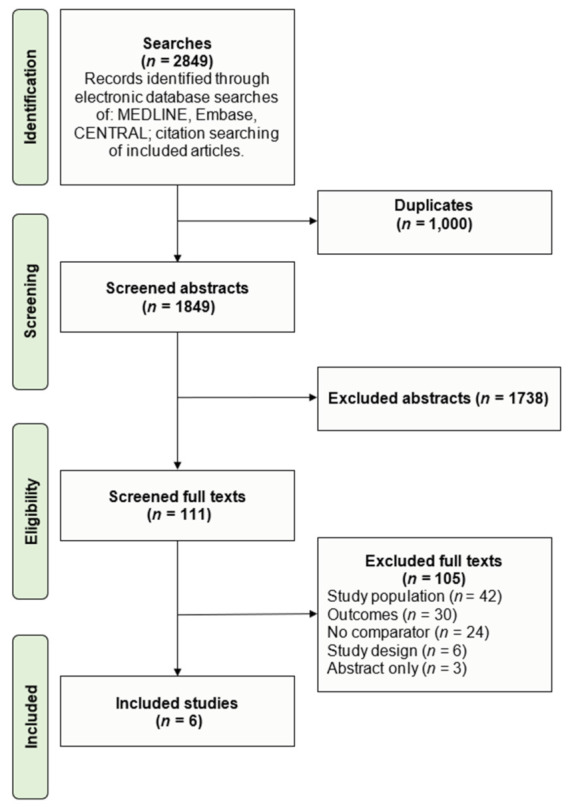
PRISMA flow diagram with study selection and reasons for exclusion.

**Figure 2 jcm-11-03955-f002:**
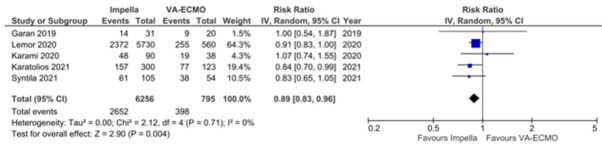
Meta-analysis of in-hospital mortality with Impella versus VA-ECMO in AMI-CS [24,25,26,27,28].

**Figure 3 jcm-11-03955-f003:**
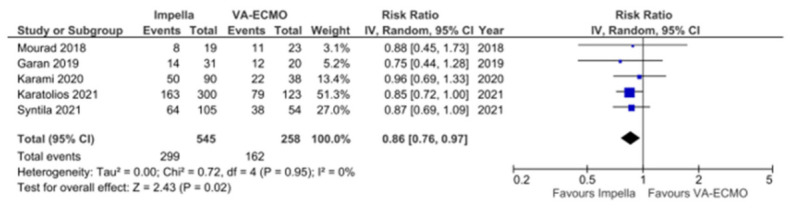
Meta-analysis of 6–12-month mortality with Impella versus VA-ECMO in AMI-CS [23,25,26,27,28].

**Table 1 jcm-11-03955-t001:** Qualitative analysis and characteristics of included studies.

Author (Year)	Study Design	Country (Time)	Population; Age in Years; % Male	Type of Impella	Ascertainment of Exposure and Outcome	Outcomes	Follow Up Period	Adjustment for Potential Confounders
Mourad (2018)	Retrospective cohort	France (January 2009–April 2015)	*n* = 42 (VA-ECMO 23 vs. Impella 19). VA-ECMO 54 (50–60) vs. Impella 57 (51–65). VA-ECMO 78% vs. Impella 89%.	Impella CP Impella 2.5 Impella 5.0	Medical records	Mortality, 30 days: VA-ECMO 7 (30%) vs. Impella 7 (37%). Mortality, 6 months: VA-ECMO 11 (48%) vs. Impella 8 (42%). Bleeding requiring surgery, in-hospital: VA-ECMO 2 (9%) vs. Impella 1 (5%). Thromboembolic events, in-hospital: VA-ECMO 2 (9%) vs. Impella 1 (5%). Peripheral limb ischaemia, in-hospital: VA-ECMO 3 (13%) vs. Impella 1 (5%).	6 months	Unadjusted. Stratified by device first used. Exclusion criteria: delayed MCS after AMI (>72 h) or after CS onset (>48 h), refractory cardiac arrest, mechanical AMI complication, aortic valve pathology, early surgical revascularization, other causes of shock.
Garan (2019)	Prospective cohort	United States (April 2015–March 2017)	*n* = 51 (VA-ECMO 20 vs. Impella 31). VA-ECMO 64 ± 9 vs. Impella 61 ± 11. VA-ECMO 80% vs. Impella 74%.	Impella CP	Prospective	Mortality, in-hospital: VA-ECMO 9 (45%) vs. Impella 14 (45%). Mortality, 12 months: VA-ECMO 12 (60%) vs. Impella 14 (45%). Peripheral limb ischaemia, in-hospital: VA-ECMO 2 (20%) vs. Impella 4 (13%).	24 months	Unadjusted. Stratified by device first used. Exclusion criteria: not stated.
Karami (2020)	Retrospective cohort	The Netherlands (2006–January 2018)	*n* = 128 (VA-ECMO 38 vs. Impella 90). VA-ECMO 55 ± 9 vs. Impella 60 ± 10. VA-ECMO 79% vs. Impella 73%.	Impella 5.0	Medical records	Mortality, in-hospital: VA-ECMO 19 (50%) vs. Impella 48 (53%). Mortality, 12 months: VA-ECMO 22 (61%) vs. Impella 50 (56%). Bleeding complications, in-hospital: VA-ECMO 5 (13%) vs. Impella 9 (10%). Peripheral limb ischaemia, in-hospital: VA-ECMO 2 (5%) vs. Impella 2 (2%).	12 months	Unadjusted. Exclusion criteria: MCS after revascularisation with CABG, IABP, Impella 2.5, cardiac arrest without ROSC.
Lemor (2020)	Retrospective cohort	United States (October 2015–December 2017)	*n* = 6290 (VA-ECMO 560 vs. Impella 5730). VA-ECMO 61 ± 12 vs. Impella 66 ± 12. VA-ECMO 83% vs. Impella 73%.	Impella CP Impella 2.5 Impella 5.0	National database, ICD-10 codes	Mortality, in-hospital: VA-ECMO (46%) vs. Impella (41%) (adjusted OR 2.34, 95% CI 1.31–4.14, *p* = 0.004). Propensity matched mortality, in-hospital (450 in each group): VA-ECMO 43% vs. Impella 27%, OR 2.10, 95% CI 1.12–3.95, *p* = 0.021).	In-hospital	Adjusted for: age, gender, race, type of AMI, comorbidities, Charlson Comorbidity Index, resuscitation status, median income, teaching status, hospital region, hospital bed size, insurance, APR-DRG Severity Classification. Exclusion criteria: Impella and ECMO, no PCI, <18 years of age, missing mortality data.
Karatolios (2021)	Retrospective cohort	Germany (September 2014–September 2019)	*n* = 423 (VA-ECMO 123 vs. Impella 300). VA-ECMO 61 ± 10 vs. Impella 69 ± 12. VA-ECMO 78% vs. Impella 76%. NB—83% VA-ECMO AMI, 87% Impella AMI. NB—cohort overlap with Syntila (2021).	Impella CP Impella 2.5	Medical records	Mortality, in-hospital: VA-ECMO 77 (63%) vs. Impella 157 (52%). Mortality, 6 months: VA-ECMO 79 (64%) vs. Impella 163 (54%). Propensity matched mortality, in hospital (83 in each group): VA-ECMO 51 (61%) vs. Impella 41 (49%). Propensity matched mortality, 6 months (83 in each group): VA-ECMO 51 (61%) vs. Impella 45 (54%). Bleeding requiring transfusion, in-hospital: VA-ECMO 21 (17%) vs. Impella 22 (7%). Peripheral limb ischaemia, in-hospital: VA-ECMO 21 (17%) vs. Impella 23 (8%).	6 months	Adjusted for: age, Charlson Comorbidity Index, vasoactive score, creatinine, pH, aetiology of shock, PaO2/FiO2, prior cardiopulmonary resuscitation. Exclusion criteria: refractory cardiac arrest in whom insertion of MCS took place under ongoing cardiopulmonary resuscitation.
Syntila (2021)	Retrospective cohort	Germany (May 2015–May 2020)	*n* = 159 (VA-ECMO 54 vs. Impella 105). VA-ECMO 62 ± 10 vs. Impella 68 ± 14. VA-ECMO NB—cohort overlap with Karatolios (2021).	Impella CP Impella 2.5	Medical records	Mortality, in-hospital: VA-ECMO 38 (71%) vs. Impella 61 (58%). Mortality, 12 months: VA-ECMO 38 (71%) vs. Impella 64 (61%). Bleeding requiring transfusion, in-hospital: VA-ECMO 19 (35%) vs. Impella 16 (15%). Peripheral limb ischaemia, in-hospital: VA-ECMO 11 (20%) vs. Impella 8 (8%). Propensity matched mortality, in hospital (40 in each group): VA-ECMO 27 (68%) vs. Impella 22 (55%). Propensity matched mortality, 12 months (40 in each group): VA-ECMO 27 (68%) vs. Impella 24 (60%).	12 months	Adjusted for: Charlson comorbidity index, vasoactive score, pH, PaO2/FiO2, lactate, first rhythm, time from collapse to ROSC. Exclusion criteria: non-AMI causes of OHCA, biventricular support.

## Data Availability

Not applicable.

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
