# Peer review of "Impella versus Venoarterial Extracorporeal Membrane Oxygenation for Acute Myocardial Infarction Cardiogenic Shock: A Systematic Review and Meta-Analysis"

_jcm, 2022, doi:10.3390/jcm11143955_

Round 1

Reviewer 1 Report

The paper regards a meta-analysis on comparison of two different mechanical circulatory support devices, Impella vs VA-ECMO for cardiogenic shock due to acute myocardial infarction. The paper is quite interesting, well written although it has some important limitations that the authors correctly mention in the discussion. Mainly there are no randomized trial and the meta-analysis is based almost exclusively on a single cohort study, whereas the other five were smaller sized studies. 

Reviewer 2 Report

The authors performed a correct systematic review and meta-analysis comparing Impella and VA-ECMO for AMI-shock. As indicated by the authors, the studies are of limited value given their observational character (high risk of bias), different inclusion criteria, use of different devices etc. My only comment is that Table 1 does not read well, since not printed in landscape. Consider revision.

Reviewer 3 Report

This paper is a systematic review and meta-analysis. Authors evaluated the in-hospital mortality after the use of mechanical circulatory support devices for acute myocardial infarction cardiogenic shock comparing VA-ECMO vs. Impella. Authors concluded that MCS with Impella was associated with a reduced risk of in-hospital and medium-term mortality as compared to VA-ECMO.

First, I would like to pay tribute to the authors for their tremendous efforts. However, the paper contains the following serious limitations.

The small number of studies (n=6) results in high heterogeneity of studies, making it difficult to obtain sufficient evidence, even for a meta-analysis.

All included studies were observational, and indications of MCS for cardiogenic shock differed significantly between IMPELLA and VA-ECMO. Only VA-ECMO, not IMPELLA, is usually indicated for cardiac arrest or biventricular heart failure. This suggests that more severe cases are more common in the VA-ECMO group.

Various causes of cardiogenic shock are mixed, including pump failure, mechanical complications, and arrhythmias, which might hinder to make a definite conclusion as to the effects of these mechanical supports on the prognosis.  

Reviewer 4 Report

In this study entitled “Impella versus Venoarterial Extracorporeal Membrane Oxygenation for Acute Myocardial Infarction Cardiogenic Shock: A Systematic Review and Meta-Analysis” the authors investigated whether the use of Impella versus VA-ECMO, in patients with AMI-CS, could affect in-hospital and 6-12 months mortality. Cardiogenic shock remains a high mortality condition despite improvement in therapies and the use of mechanical circulatory supports. In this scenario, many unmet needs remain to be explored and, if on one hand studies are required to investigate the best therapeutic strategy, on the other hand the critical status of patients with CS and the urgency of the actions usually required to support them make it difficult to have good quality prospective (and randomized) trials. This turns into a lack of high level evidence and therefore, when performing a metanalysis, this aspect has to be taken into account and recognized. 

Said that, the article by Batchelor included many patients and is well written, and therefore is of potential interest. I however suggest major revisions to improve its quality and to let the readers better understand the clinical implications of their results, avoiding to risk of suggesting the routine use of IMPELLA over VA-ECMO in AMI-CS. Here are my comments:

1.     IMPELLA and VA-ECMO are both MCS devices, but they are designed for and have different indications. In section introduction the authors should specify what are the indications for each.

2.     Please include a detailed description, where possible, about which were the indications of choice for IMPELLA rather than ECMO in the included studies. Patients treated with VA-ECMO have usually more severe forms of cardiogenic shock, so that a higher mortality would be expected. This could have affected the results of the study and has to be clearly stated and discussed.

3.     The authors should in general temperate their conclusions because there were several possible biases in study and patient selection, the studies were almost all retrospective and observational and, to date, a routine use of IMPELLA over ECMO cannot be recommended (and would be dangerous). This study can be considered hypothesis generating and can be helpful to design prospective randomized trials in the future. 

4.     I suggest adding a further sensitivity analysis reporting the exclusion of each single study at time, to assess whether one of those influenced the results of the analysis. Then comment in the discussion.

5.     The authors, in section methods, should report all the complete and detailed search strategy, with the words included in the database search and doing this for each database screened. Moreover, the supplement 1 is not very clear and it not easy to understand how many records were identified and then included in the analysis.

6.     In the abstract the authors should report the definition of not only the primary outcome, but also the secondary outcomes.

7.     Introduction: “To date, there have been few randomised trials comparing Impella and VA-ECMO in AMI-CS”. This sentence requires a reference.

8.     The Figure 1 and, most importantly, Table 1 have been probably completely changed during PDF generation of the article, so that they are not understandable. Please, make sure that will not occur when resubmitting them.

Round 2

Reviewer 3 Report

The revised version met my questions well.

Reviewer 4 Report

The authors provided a revised version of the manuscript, addressing most of the comments. The resulted in an improvement of the quality of the manuscript and the conclusions result now more temperated and supported by their results.